# The added value of cognition-targeted exercise versus symptom-targeted exercise for multiple sclerosis fatigue: A randomized controlled pilot trial

Azza Alketbi[1], Salah Basit[2], Nouran Hamza[3], Lori M. Walton[4], Ibrahim M. Moustafa[1,2]*

1 Department of Physiotherapy, College of Health Sciences, University of Sharjah, Sharjah, United Arab Emirates, 2 Faculty of Physical Therapy, Cairo University, Giza, Egypt, 3 Biostatistician at Medical Agency for Research and Statistics (MARS), Egypt, 4 Department of Physical Therapy, University of Scranton, Scranton, Pennsylvania, United States of America

* labuamr@sharjah.ac.ae

## Abstract

### Background

Fatigue is considered one of the most common symptoms of multiple sclerosis (MS) and lacks a current standardized treatment. Therefore, the aim of this study was to examine the feasibility and effectiveness of a cognition-targeted exercise versus symptom-targeted exercise for MS fatigue.

### Methods

In this Pilot, parallel-group, randomized controlled trial, sixty participants with multiple sclerosis, were randomly assigned to either a Cognition-Targeted Exercise (CTE) (N = 30, mean age 41) or a Symptom-Targeted Exercise (STE) (N = 30, mean age 42). The participants in the experimental group received eight, 50-minute sessions of weekly Cognitive Behavior Therapy (CBT) in addition to a CTE Program; whereas, participants in the control group received eight, 50-minute sessions of weekly CBT in addition to the standardized physiotherapy program (STE Program). Feasibility was assessed through recruitment rate, participant retention, adherence and safety, in addition to clinical outcome measures, including: (1) Modified Fatigue Impact Scale (MFIS), (2) Work and Social Adjustment Scale (WSAS), (3) Hospital Anxiety and Depression Scale (HADS), and Perceived Stress Scale (PSS). All outcome measures were assessed at baseline (pretreatment), following completion of the eight visit intervention protocol, and at 3-months follow-up.

### Results

The recruitment rate was 60% and 93% of participants completed the entire study. The recruited participants complied with 98% of the required visits. No adverse events were recorded. A Generalized Estimation Equation Model revealed a significant difference over

**Data Availability Statement:** All relevant data are within the paper and its Supporting information files.

**Funding:** The authors received no specific funding for this work.

**Competing interests:** The authors have declared that no competing interests exist.

time as an interaction term during the post and follow up visit for all clinical outcome measures (p < .001).

## Conclusion

The addition of CTE to CBT exhibited positive and more lasting influence on MS fatigue outcomes compared to Symptom-Targeted Exercise (STE). Feasibility and efficacy data from this pilot study provide support for a full-scale RCT of CTE as an integral component of Multiple Sclerosis fatigue management.

## Introduction

Multiple Sclerosis (MS) is a chronic demyelinating disease of the central nervous system and is broadly known as a common neurologic complication in humans [1]. It is estimated that over 2.5 million people are suffering from this ailment all around the world [2]. It is most prevalent among females, 20 to 40 years of age [3]. The cause has not yet been detected and no definite treatment has been found; however, pharmacological interventions have exhibited some success in the reduction of intensity and recurrence of the disease process [4]. MS represents a wide range of impairments, including: muscle pain and cramping, insentience, cognitive, sensory, vision and speech dysfunction, gastrointestinal excretory problems and fatigue [5]. Complaints of fatigue for people with MS are reported at 76–97% [1]. The causes of fatigue in people with MS have not been clearly defined in the research [6]. Approximately 65% of people with MS reported fatigue-related symptoms as one of the three most bothersome aspects of their symptoms [7], with a profound side effect on health-related quality of life and daily activity performance [8]. Fatigue also interfered with daily activities for people with MS, causing numerous troubles in occupational, educational, economic, recreational and family areas, ultimately producing negative outcomes in social and personal communications and mental health outcomes [9].

Several published reviews have examined the effectiveness of individual types of fatigue management interventions for people with MS. Current review of evidence for pharmacologic intervention trials targeting fatigue in people with MS have shown weak and inconclusive results [7, 10]. In comparison, recent reviews of exercise training [11] and cognitive behavioral therapy (CBT) [12] suggest that these interventions may be beneficial for MS fatigue management.

The body of research investigating the effect of CBT and exercise interventions is expanding, but to our knowledge, there are insufficient studies that examine the combined effect of CBT and exercises to target fatigue for people with Multiple Sclerosis.

While the role of CBT in contributing to the reduction of fatigue severity is well established [13–15], there continues to be controversy regarding the role that physical activity and exercise play in fatigue management [16–21]. No single, optimal exercise modality has been established to address MS fatigue. Furthermore, most studies show variability in subtype exercise choice, depending on specific MS symptoms, level of functional ability and therapist preference. There is a lack of clarity in identification of specific type and components of exercise that should be included in an intervention to achieve positive fatigue management benefits for people with MS [22].

In light of current research findings, we cannot find a clear answer to the question of whether different types of exercise may affect fatigue differently. The purpose of the present

study was to determine whether the treatment effect of CBT on fatigue is mediated by adding different exercise approaches (Cognition-Targeted Exercise versus symptom-contingent exercises). Related aims included feasibility outcomes related to rate of recruitment, participant retention and/or completion rate, adherence to treatment allocation and assurance of protocol safety. The results of this study are an important addition to the validation of CBP treatment approaches and interventions that rely upon physical exercise and activity programs in some way to reduce fatigue in people with Multiple Sclerosis.

## Methods

### Study design and setting

This study was a prospective, assessor-blinded, parallel-group, randomized controlled clinical trial, comparing one group that received CBT plus Cognition-Targeted Exercise (CTE) to a control group receiving CBT plus Symptom-Targeted Exercise (STE). The setting was Farouk Hospital—Egypt. This trial was registered with the Clinical-Trials.gov NCT04699370 where the full protocol can be accessed (https://register.clinicaltrials.gov/prs/app/action/SelectProtocol?sid=S000AJQR&selectaction=Edit&uid=U0005FZM&ts=2&cx=-g5ewh2). The study protocol was approved by the local ethics committee. Informed consents were provided to and obtained from all participants prior to data collection in accordance with relevant guidelines and regulations.

(The protocol for this trial and supporting CONSORT checklist are available as supporting information; see S1 Checklist and S1 File).

### Sample size

Priori sample size calculation indicated that 100 patients per each group were required to detect a minimal clinically important change of 4 points on the MFIS [23] (assuming a level of significance of 5% and a power level of 90%). As a pilot study we used 25 patients per each group and to account for possible participant dropouts, the sample size was increased by 20%.

### Participants

Both male and female subjects were included in this study if they aged between 20–50 years, have relapsing remitting or progressive MS and met the following criteria: (1) Diagnostic criteria for MS were confirmed by a neurologist [24]. (2) Being within normal or average dysfunction and excluding those scoring $\geq 6$ in the Expanded Disability Status Scale (EDSS); (3) Being identified as a case level of fatigue; fatigue score of 4 or greater on the fatigue Scale [25]. Potential participants were excluded from the study if they had any serious psychiatric disorders including major affective disorder or any chronic illness that may affect their fatigue level.

### Allocation and concealment

An independent person blinded to group allocation and not involved in any other aspect of the study performed the randomization in a 1:1 ratio utilizing a method of random permuted blocks of different sizes. A permuted block randomization sequence was generated using a random number generator (www.randomizer.org). Each block was placed in a sequence of consecutively numbered sealed envelopes. Sequentially numbered and sealed opaque envelopes containing the sequence were stored in a locked drawer. For each enrolled participant, the study coordinator (not involved in the preparation of the allocation sequence) retrieved and opened the next sequentially numbered envelope and assigned the participant according to the random allocation scheme.

## Intervention protocol

The participants in the experimental group received eight 50-minute sessions of weekly CBT based on van Kessel's model [26] in addition to a Cognition-Targeted Exercise CTE program; whereas, participants in the control group received eight 50-minute sessions of weekly CBT in addition to a standardized physiotherapy program, consisting of Symptom-Targeted Exercises (STE).

- CBT was designed on the basis of van Kessel's Model. The main objective of this treatment is to challenge all external factors (e.g. behavioral, cognitive, and affective factors) expected to participate in the development and persistence of fatigue in MS patients. The treatment sessions were delivered on an individual basis. Table 1 summarizes the session contents.

- For Cognition-Targeted Exercise (CTE), all standardized physical therapy exercises were performed in a time contingent rather than in a symptom-contingent way. Goal setting is essentially done together with the patient, focusing on functionality instead of fatigue relief. Progression to the next level was preceded by mentally visualizing the task for enhancement of successful execution of the targeted movement. During the CTE training, the therapist placed great emphasis on patient's cognitive report of their problems, so that patients will have positive perceptions regarding their illness and treatment outcome. In addition, the patient's perceptions about each exercise and anticipated consequences of the exercises were discussed during the session.

- The standardized physical therapy protocol consisted of eight, half-hour, individualized physiotherapy sessions, over a 4-week period. This program consisted of twice-weekly supervised general aerobic, strengthening and flexibility exercise sessions based on symptom-targeted contingencies. This exercise program is typically implemented in routine clinical practice.

All intervention procedures for the study and control group were delivered individually, face to face, by the same physiotherapist, with 10 years prior experience, who received specialized training and certification in the techniques employed in this study to minimize inter-therapist variation, enhance fidelity and to mimic a clinical oriented patient-physiotherapist

**Table 1. Summary of CBT sessions.**

| First session | Overview of causes of MS fatigue; explanation of cognitive behavioral model for MS fatigue. |
|---|---|
| Second session | Introduction of treatment rationale, which includes an explanation of CBT and how it relates to MS fatigue |
| Third session | Education on how patterns of rest and activity or over-activity affect the body and fatigue. |
|  | Participants are encouraged to set goals to improve levels of resting, activity, and exercise to set goals to improve levels of resting, activity, and exercise |
| Fourth session | Information is provided on sleep patterns (sleeping too much or too little) and impact on fatigue and behavioral techniques (basic sleep hygiene) |
| Fifth session | The concepts of symptom focusing and symptom attribution are introduced, and how these can have an impact on MS fatigue. Alternative explanations of somatic symptoms are discussed, |
| Six session | The concept of negative thoughts is introduced and impact on fatigue and mood. |
| Seventh session | Basic stress management and coping with emotions is discussed. Participants are encouraged to set goals for stress management and practicing alternative ways to manage their emotions |
| Eighth session | The importance of social support for MS patients is discussed, and participant's personal support systems are reviewed. Participants are encouraged to continue to employ the skills they have learned throughout the manual to manage their fatigue |

relationship. Compliance with the exercise program was calculated as total number of session's attended/total number of sessions available.

## Outcome measures

Multiple outcome measures were collected at baseline (pretreatment), the next day following the completion of the eight visit intervention, and at 3-months follow-up. The order of measurements was the same for all participants. Modified fatigue impact scale (MFIS) was the primary treatment outcome to determine the fatigue level. The MFIS has been recommended by "the fatigue guidelines development panel of the MS Council for Clinical Practice Guidelines", as the main outcome measures for assessing MS related fatigue [27, 28].

**Secondary outcome measures included**:

- **The Work and Social Adjustment Scale**. Fatigue-related impairment was assessed using a reliable and valid Work and Social Adjustment Scale [29].

- **Hospital Anxiety and Depression Scale**. Improvements in mood was assessed using a reliable and valid Hospital Anxiety and Depression Scale [30, 31], a commonly used self-report instrument for detecting states of depression and anxiety in patients with medical illnesses.

- Finally, the perception of stress. Was measured using a valid and reliable **Perceived Stress Scale** [32, 33].

The main feasibility outcomes included recruitment rate, retention or completion rate, adherence rate and safety. Recruitment rate was a simple ratio of the number of identified participants vs. those who actually agreed. Whereas, the completion rate was indicated by the number of participants that completed the entire study. Adherence rate was quantified by the number of treatments made vs. the total recommended. For safety assessment, the number and nature of adverse events were recorded on a weekly basis during the intervention period and at every month during the follow up period. All participants were assessed by two physiotherapists, who have at least 10 years' experience in neurorehabiliation and were not the same therapist who provided the intervention protocol. Assessors were blinded to group allocation to prevent any bias.

## Data analysis

The statistical procedure depended on the principle of intention to treat for between-groups comparisons. The normal distribution of all descriptive statistics baseline variables was determined using the Kolmogorov–Smirnov test; where continuous data is noted as mean with standard deviation (SD) in the text and tables. Equality of variance was assessed with Levene's test; attaining a 95% confidence level, p-value > 0.05. To follow up and compare the effects of the 2 alternative treatments over 3 months, we examined the results using a generalized Estimation Equation (GEE) Model. The statistical analysis was performed using the SPSS statistical software system The SPSS Statistics 26 ® (IBM Corp.) and RStudio IDE. (RStudio, Inc.).

## Results

The experimental and control groups were similar for age, gender, EDSS score and all outcome measures. The baseline participant demographics of all patients are shown in Table 2.

**Table 2. Baseline participant demographics.**

|  | Cognition-targeted group (n = 30) | Symptom-targeted group (n = 30) |
|---|---|---|
| Age(y) | 41 ± 6 | 42 ± 4 |
|  | Range 47–65 | Range 45–64 |
| Weight(kg) | 59 ± 9 | 60 ± 8 |
| Gender (%) |  |  |
| Male | 5 | 4 |
| Female | 25 | 26 |
| EDSS score | 4.5 | 4.3 |
| Marital status |  |  |
| Married | 22 | 20 |
| Single | 1 | 2 |
| Widow /divorced | 7 | 8 |
| Employment status |  |  |
| Working | 27 | 28 |
| retired | 3 | 2 |
| Educational level |  |  |
| University or higher | 19 | 18 |
| High school | 2 | 1 |
| Junior high school or less | 9 | 11 |
| Disease duration, years, mean (SD) | 12±4 | 11±3 |
| Disease course (RRMS/SPMS), | 28/2 | 29/1 |
| Medication at enrolment |  |  |
| Interferon | 6 | 8 |
| Fingolimod | 15 | 12 |
| Natalizumab | 9 | 10 |

EDSS: Expanded Disability Status Scale. RRMS: relapsing remitting multiple sclerosis; SPMS: secondary progressive multiple sclerosis; SD: standard deviation.

## Feasibility

**Recruitment.** The exclusion criteria for this study removed 40% of individuals initially interested in participation, resulting in a recruitment rate of 60%. One hundred participants were initially screened and 60 subjects were found to be eligible to enroll in the study. In total, 60 (100%) participants completed the first follow-up at 4 weeks of treatment and 56 (93%) completed the entire study including the 3-month follow-up after the four week intervention. A diagram of participant retention and randomization throughout the study is shown in Fig 1. The study adherence rate was relatively consistent for both groups. Ninety-eight percent of participants in the CTE group attended 100% of the intervention sessions. In the STE group, 95% of participants attended 100% of sessions. Participant illness accounted for 98% and illness in a family member accounted for 2% of missed appointments. No adverse effects were identified during the 12-week intervention period. Only one participant was hospitalized for 3 days because of a generalized infection unrelated to the study.

A generalized Estimation Equation (GEE) Model was developed and used to examine the influence of CTE over STE on changes in Modified Fatigue Impact scale (MFIS), Work and social Adjustment scale (WSAS), Hospital Anxiety Scale (HAS) and Perceived Stress Scale (PSS). Pre-treatment scores were considered as reference values for estimating the post and

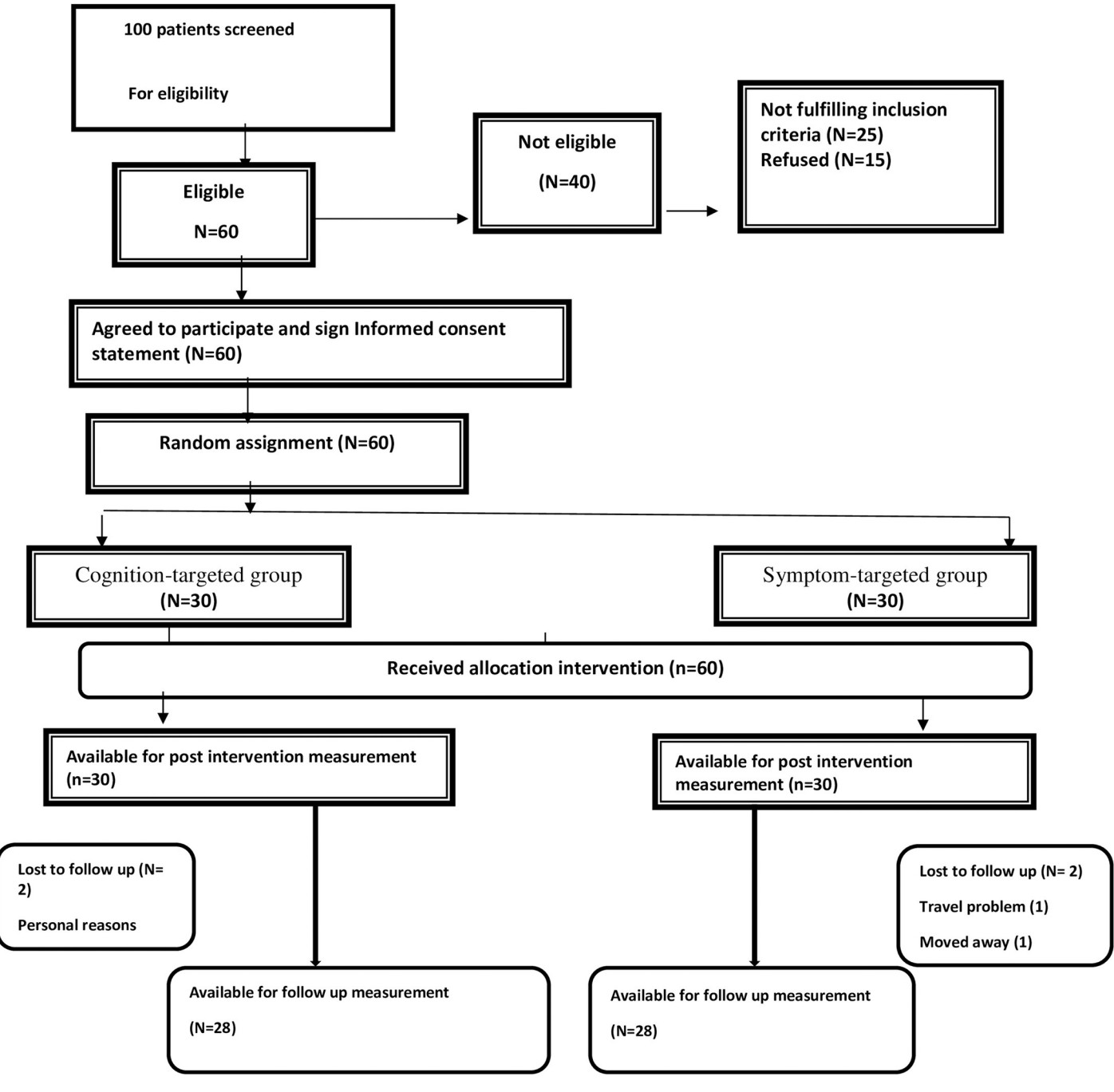

**Fig 1. Flow chart.**

follow up changes among groups. Models selected were chosen according to the least quasi-likelihood under the independence model criterion.

A Generalized Estimation Equation Model revealed a significant difference over time as an interaction term during the post and follow up visit for. Regarding the Hospital Anxiety Scale (HAS), an increase by 0.32 units was reported as a main effect for the CTE compared to STE group. However this observed effect was found to be reduced significantly over the post and

follow up visits in response to CTE by -3.96 and -4.45, respectively. There was also a non-significant decrease by -1.08 units as a main effect of CTE, compared to STE group. This decrease remained consistent with the null hypothesis even after considering the time factor for the post treatment visit. However, the interaction effect of CTE at the follow up visit showed a significant decrease by -4.616 units. (The data are available as supporting information; see S1 Data).

## Discussion

This randomized controlled trial compared the outcomes of fatigue, work and social adjustment, hospital anxiety, depression, and perceived stress scale between a group that received the Cognition-Targeted Exercise (CTE) coupled with Cognitive Based Therapy (CBT) to that of a group that received Symptom-Targeted Exercise (STE) coupled with CBT. The adherence to treatment assignment during the 4-week, intervention period and compliance with follow-up were acceptable, with no serious adverse events. The between group analysis revealed a significant difference over time as an interaction term during the post and follow up visit. In the cognition-targeted exercise group, the amount of Modified Fatigue Impact Scale (MFIS) and other management outcomes related to work and social adjustment, hospital anxiety and perceived stress improved significantly and this improvement was maintained at the 3-month follow-up. In contrast, the STE group revealed significantly less differences in fatigue level and other management outcomes between baseline and 4-weeks of treatment, and at the 3-month follow-up fatigue scores increased again (Table 3).

To our knowledge, there is no study that has evaluated the comparative effectiveness of cognition-targeted exercise and symptom-targeted exercise. So far, few existing studies have compared the effects of different exercise approaches that have heterogeneous results [34]. There is a substantial body of research showing that the level of physical activity is an effective component in chronic fatigue management for patients with MS [16, 35]. However, according to Wiborg et al. [13], the effect of CBT on fatigue "is not mediated by a persistent increase in physical activity". The results of our study are also consistent with earlier research on CBT for chronic fatigue in which a reduction in fatigue was associated with a change in illness beliefs and not the physical activity level [20, 21].

Diverse results related to exercise approach in patients with MS found in comparative literature may be one explanation for the lack of consensus for interventions for reducing fatigue. Previous studies have challenged the behavioral, and cognitive factors that play a role in the development and persistence of fatigue in MS patients. Other studies focused only on the physical fitness status. The majority of previous studies investigated non-fatigued groups of patients with MS [34], thus the level of evidence from this group of studies may be limited and not generalizable to patients with MS and fatigue [36–40].

However, no clear explanation is available for the superiority of CTE, the plausible mechanism behind the positive effect on MS fatigue, which is based on the theory that excessive physiological motor fatigue is mainly central in origin, rather than a consequence of intramuscular changes. It, logically, follows that when patients learned of their ability to increase their level of physical activity, despite their symptoms, the belief of having little control over their condition may have, also, changed [13]. This explanation is also consistent with other studies that highlighted the role of changing the illness belief as an important factor in fatigue reduction [21, 41]. In light of the findings of this study, changing illness-related cognitive behaviors may also play a more crucial role in CBT for Chronic Fatigue Syndrome (CFS) compared to a focus on sole increase in physical activity.

**Table 3. GEE model for unit change from baseline to 4- week post treatment and the 3-month follow-up, cognition-targeted.** Exercise versus symptom-targeted exercise for multiple sclerosis fatigue.

| Modified Fatigue Impact | GEE coefficient | 95% confidence interval | Standard error | Wald | p-value |
|---|---|---|---|---|---|
| (Intercept) | 49.6 | 47.49 To 51.71 | 1.08 | 2117.9 | <0.001 |
| Cognition-targeted | -2.16 | -6.04 To 1.72 | 1.98 | 1.19 | 0.2747 |
| Post | -6.32 | -8.93 To -3.71 | 1.33 | 22.47 | <0.001 |
| Follow | -4.3 | -7.44 To -1.15 | 1.61 | 7.15 | <0.001 |
| Cognition-targeted: post | -12.88 | -17.31 To -8.45 | 2.26 | 32.4 | <0.001 |
| Cognition-targeted: follow | -15.32 | -20.3 To -10.34 | 2.54 | 36.35 | <0.001 |
| Work and Social Adjustment Scale | GEE coefficient | 95% confidence interval | Standard error | Wald | p-value |
| (Intercept) | 18.28 | 17.15 1 To 9.408 | 0.576 | 1008.74 | <0.001 |
| Cognition-targeted | -1.36 | -2.91 To 0.186 | 0.789 | 2.97 | 0.08468 |
| Post | -2.96 | -4.69 To -1.231 | 0.882 | 11.26 | 0.00079 |
| Follow | -2.367 | -4.17 To -0.561 | 0.921 | 6.6 | 0.01020 |
| Cognition-targeted: post | -3.68 | -5.82 To -1.537 | 1.094 | 11.33 | 0.00076 |
| Cognition-targeted: follow | -5.162 | -7.40 To -2.923 | 1.142 | 20.41 | <0.001 |
| Hospital Anxiety | GEE coefficient | 95% confidence interval | Standard error | Wald | p-value |
| (Intercept) | 15.08 | 14.16 To 16 | 0.47 | 1031.12 | < 2e-16 |
| Cognition-targeted | 0.32 | -0.858 To 1.498 | 0.601 | 0.28 | 0.59449 |
| Post | -1.72 | -3.108 To -0.332 | 0.708 | 5.9 | 0.01517 |
| Follow | -1.863 | -2.922 To -0.803 | 0.541 | 11.87 | 0.00057 |
| Cognition-targeted: post | -3.96 | -5.682 To -2.238 | 0.879 | 20.31 | <0.001 |
| Cognition-targeted: follow | -4.45 | -5.967 To -2.933 | 0.774 | 33.06 | <0.001 |
| Perceived Stress | GEE coefficient | 95% confidence interval | Standard error | Wald | p-value |
| (Intercept) | 22.12 | 20.36 2 To 3.878 | 0.897 | 608.41 | <0.001 |
| Cognition-targeted | -1.08 | -3.48 To 1.321 | 1.225 | 0.78 | 0.3779 |
| Post | -2.64 | -5.01 To -0.266 | 1.211 | 4.75 | 0.0293 |
| Follow | -1.772 | -4.51 To 0.961 | 1.394 | 1.62 | 0.2038 |
| Cognition-targeted: post | -3.04 | -6.43 To 0.348 | 1.729 | 3.09 | 0.0786 |
| Cognition-targeted: follow | -4.616 | -8.05 To -1.179 | 1.753 | 6.93 | 0.0085 |

Symptom-targeted Exercise was the reference value.

CBT is based upon the principle that physical, behavioral, cognitive, and affective responses and functions interact with one another with mutual effects [1]. Therefore, the addition of CTE, which focuses on the patient's awareness of the connection between body and cognitive belief systems, may contribute toward a change in other systems (e.g. affective and physical response), resulting in a greater reduction in symptoms of fatigue for patients with MS, than those achieved by CBT, alone.

## Study limitations and strategies for improvement

Our study strengths include successful blinding of the outcome assessors, a limited loss to follow-up, and high compliance rate for both of our groups. However, we propose several limitations of this study, pointing to necessary future research work on this topic. First, our project only included a short-term follow-up of 3-months after intervention was terminated. It is unclear how long the identified fatigue related changes would remain and whether patient compliance may also be an issue. Second, because we chose participants with a scoring of less than six in the expanded disability status scale to investigate, it is unknown how our study

results might relate to people with MS who have higher levels of disability. Third, within the resource constraints of a pilot investigation, only a small sample size was feasible. While our study had statistically significant findings and raised the possibility of Cognition-Targeted Exercise CTE benefits, no reliable conclusions about mood effects of CTE can be drawn from such a small sample. As Button et al. [42] emphasize, "a study with low statistical power reduces the likelihood that a statistically significant result reflects a true effect." Thus, replication studies with larger sample sizes are critical to further evaluating any potential mood effects of this type of exercise. Despite of the above mentioned limitations, the present study can be regarded as a positive step toward finding and adopting non-pharmacological interventions to ameliorate fatigue levels in patients with Multiple Sclerosis.

## Conclusion

The addition of the Cognition-Targeted Exercise (CTE) to Cognitive-Behavioral Therapy (CBT) revealed positive and more lasting influence on multiple sclerosis management outcomes compared to those received CBT with Symptom-Targeted Exercises (STE). Feasibility and efficacy data from this pilot study provide support for a full-scale RCT based on CTE as an integral component of Multiple Sclerosis fatigue management.

## Supporting information

**S1 Checklist. CONSORT 2010 checklist of information to include when reporting a randomised trial.**
(DOC)

**S1 File.**
(PDF)

**S1 Data.**
(XLSX)

## Author Contributions

**Conceptualization:** Ibrahim M. Moustafa.

**Data curation:** Azza Alketbi, Nouran Hamza.

**Formal analysis:** Salah Basit, Nouran Hamza, Ibrahim M. Moustafa.

**Investigation:** Azza Alketbi, Salah Basit.

**Methodology:** Azza Alketbi, Salah Basit.

**Project administration:** Ibrahim M. Moustafa.

**Supervision:** Lori M. Walton, Ibrahim M. Moustafa.

**Validation:** Nouran Hamza, Lori M. Walton.

**Visualization:** Lori M. Walton.

**Writing – original draft:** Azza Alketbi, Salah Basit, Ibrahim M. Moustafa.

**Writing – review & editing:** Nouran Hamza, Lori M. Walton, Ibrahim M. Moustafa.

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
