## [Decision Letter · Decision Letter 0]

1 Jul 2021

PONE-D-21-18476

The Added Value of Cognition-targeted Exercise Versus Symptom-targeted Exercise for Multiple Sclerosis Fatigue: Randomized Controlled Trial

PLOS ONE

Dear Dr. Moustafa,

Thank you for submitting your manuscript to PLOS ONE. After careful consideration, we feel that it has merit but does not fully meet PLOS ONE’s publication criteria as it currently stands. Therefore, we invite you to submit a revised version of the manuscript that addresses the points raised during the review process.

We look forward to receiving your revised manuscript.

Kind regards,

Walid Kamal Abdelbasset, Ph.D.

Academic Editor

PLOS ONE

Journal Requirements:

3. Please amend your manuscript to include your abstract after the title page.

Additional Editor Comments (if provided):

Reviewers' comments:

Reviewer's Responses to Questions

**Comments to the Author**

1. Is the manuscript technically sound, and do the data support the conclusions?

Reviewer #1: Partly

Reviewer #2: No

Reviewer #3: Yes

2. Has the statistical analysis been performed appropriately and rigorously? 

Reviewer #1: I Don't Know

Reviewer #2: No

Reviewer #3: Yes

3. Have the authors made all data underlying the findings in their manuscript fully available?

Reviewer #1: Yes

Reviewer #2: Yes

Reviewer #3: Yes

4. Is the manuscript presented in an intelligible fashion and written in standard English?

Reviewer #1: Yes

Reviewer #2: No

Reviewer #3: No

5. Review Comments to the Author

Reviewer #1: Important note: This review pertains only to ‘statistical aspects’ of the study and so ‘clinical aspects’ [like medical importance, relevance of the study, ‘clinical significance and implication(s)’ of the whole study, etc.] are to be evaluated [should be assessed] separately/independently. Further please note that any ‘statistical review’ is generally done under the assumption that (such) study specific methodological [as well as execution] issues are perfectly taken care of by the investigator(s). This review is not an exception to that and so does not cover clinical aspects {however, seldom comments are made only if those issues are intimately / scientifically related & intermingle with ‘statistical aspects’ of the study}. Agreed that ‘statistical methods’ are used as just tools here, however, they are vital part of methodology [and so should be given due importance].

COMMENTS: The effect size is taken from a pilot study with total n=10 (i.e., 5 subjects in each group assuming 1:1 allocation ratio, which is too small to yield any useful/meaningful statistic) [Sample size section (line 163-7): The effect size estimate regarding the primary outcome MFIS, was obtained from a pilot study in which 10 participants underwent a similar protocol. The effect size for the differences between the groups was estimated to be 0.79. Accordingly, given a significance level of 5% and statistical power of 80%, 25 patients for each treatment arm were needed. To compensate for potential attrition, we increased the sample size by 20%.giving a total number of 30 patients per group]. The effect size used to estimate required sample size seems to be very large {unreasonably large}. In such (psychological therapy) studies [The objective of this study was to explore the added value effect of cognition-targeted exercise versus symptom-targeted exercise for MS fatigue], the effect size achieved/attained/seen is generally (in most cases) is of small to ‘medium’ level.

By referring to table-2 on page 158 of J. Cohen’s paper “A power primer” in Psychological Bulletin, 1992, vol.:112, pp 155-159 even for medium effect size you need n=64 per group (type-I error=0.05, power=80%).

{Please note that the ‘effect size’ assumed should have some valid basis (exact published reference needs to be quoted) &/or reasonable/realistic, else the study is very likely ‘not to be able to’ detect a difference despite its presence}

Note that ‘The general linear model with repeated measures’ is not a technique originally developed (for testing interaction terms it is definitely nice, however, not) for between groups comparison. Direct between groups comparison techniques are available. Remember that though the measures/tools used are appropriate [e.g. modified fatigue impact scale (MFIS), work and social adjustment scale, hospital anxiety and depression scale, perceived stress scale] most of them yield data that are in [at the most] ‘ordinal’ level of measurement [and not in ratio level of measurement for sure {as the score two times higher does not indicate presence of that parameter/phenomenon as double (for example, a Visual Analogue Scales VAS score or say ‘depression’ score)}]. Then application of suitable non-parametric test(s) is/are indicated/advisable [even if distribution may be ‘Gaussian’ (i.e. normal)]. Agreed that there is/are no non-parametric test(s)/technique(s) available to be used as alternative in all situation(s) [suitable / most desired/applicable], but should be used whenever/wherever they are available. [Assumptions of ‘general linear model with repeated measures’ are unlikely to be fulfilled making their use invalid]

What is in lines 174-5 (we examined the results using 2 way repeated measures anova) is not reflected properly in tables [2 & 3]. Why there are two ‘P’ values [one each in column 4 & 5, row 4]. Application of repeated measures ANOVA yields only one ‘P’ (for each parameter/variable), in my knowledge. Then which test it is? If it is done after getting a significant ‘F’ by repeated measures ANOVA, then is not that “multiple testing” which requires adjustment in ‘P’? What the CI [in column 4 & 5, row 4] is for? Is it for difference in means? What is the interpretation? Moreover, one should use ‘to’ in/while presenting CI {example, [-5.9 -3.6] should be [-5.9 to -3.6]}.

It is well-known that while reporting [findings from] ‘Clinical Trial’ follow CONSORT guidelines. Even word ‘CONSORT’ or important items {like Random Sequence generation (Item 8a), Allocation concealment (Item 9)} of/in CONSORT checklist are not found [since your article type is ‘Clinical Trial’, you are supposed to cover these items in the report].

Two limitations of this study pointed out in lines 254-7 are not (unfortunately) the only ones. As pointed out in ‘important note’ above “This review pertains only to ‘statistical aspects’ of the study and so ‘clinical aspects’ should be assessed separately/independently. In my opinion, to rescue this article (which is quite possible), lot of re-vision is needed. Therefore, I have to unfortunately recommend ‘major revision’.

Reviewer #2: Title

What did you need in the title

Discuss What do you mean here by added value?

The CBT used too much and published before, what was the new in your study

What is your question and answer here?

Abstract:

The background needs to be shortened.

Methods section is poorly framed. It has to be re-written.

Demographic profile of patients is not mentioned. P value are all .0005 – please check again

poorly Key words and not illustrative (type of technique or therapy you used )

Introduction:

1. Explain the rationale of the study. Please delete information unrelated to objective so that the section is short and sweet. Kindly focus on three elements of introduction.

a. What is known about the topic? (Background)

b. What is not known? (The research problem)

c. Why the study was done? (Justification)

2. Objective is not clear as mentioned above.

Write on

patients with multiple sclerosis

cognitive behavioral model of MS fatigue

Symptom-targeted Exercise

objective of the current study

Methods

Totally unclear

Rewrite by these consequences

-Study design, setting, sample size, Participant (inclusion and exclusion criteria),

-Which stage of MS you select –mention if the patients have any symptoms or remission –the patient under treatment during the study (mention all with its side effect )

-Intervention (explain CBT)

- Comparison. Ethics and end point

LINE 101-102

psychological disorders or any chronic illness that may affect their fatigue

discus what are you mean here

LINE 103

A third person blinded to group allocation

Where were the 1st and 2nd person

Line 110- 111 (explain CBT )

The subjects in the experimental group received eight 50-minute sessions of weekly CBT based on van Kessel’s model

CONSORT 2010 checklist need revision pages not accurate for each element

13a and 13b not covered in page 8

Page 8 include (data analysis and results )

Results

P value are all P<0.0005 in the 1st paragraph page 9 and 0.001 in the tables clarify please (why same results for all values )

Also in table 1 gender (numbers not clear )30-25 v 29-26

I think statistics need readjustment completely

Discussion:

The discussion section needs to be written and described scientifically after correction needed for results

References

Rewrite again

Most reference are old like (5,8,13,14,18,20,30) and not completed (volumes and issues ) like (33,34,35,29,26,15,1)

Reviewer #3: This work explores a novel target for the treatment of fatigue in patients with MS, and I congratulate the authors for their efforts to carry out this trial.

As additional data, it would be interesting to know the proportion of patients with different MS phenotypes recruited in the study, and if there were differences in the response to therapy in the different clinical forms.

In addition, it would be interesting to know if the patients who intervened were under immunomodulatory treatment, and what type of treatment, since we know that interferons can cause fatigue

Finally, an English correction by a native speaker would be recommended.

6. PLOS authors have the option to publish the peer review history of their article (what does this mean?). If published, this will include your full peer review and any attached files.

Reviewer #1: No

Reviewer #2: No

Reviewer #3: No

---

## [Author Response · Author response to Decision Letter 0]

26 Aug 2021

Comments 

 Comments Please ensure that your manuscript meets PLOS ONE's style requirements, including those for file naming. The PLOS ONE style templates can be found at https://journals.plos.org/plosone/s/file?id=wjVg/PLOSOne_formatting_sample_main_body.pdf and https://journals.plos.org/plosone/s/file?id=ba62/PLOSOne_formatting_sample_title_authors_affiliations.pdf

Responses Done as suggested

Comments We note that you have indicated that data from this study are available upon request. PLOS only allows data to be available upon request if there are legal or ethical restrictions on sharing data publicly. For information on unacceptable data access restrictions, please see http://journals.plos.org/plosone/s/data-availability#loc-unacceptable-data-access-restrictions.

Response There is now restriction and we have uploaded the data file

Comments Please amend your manuscript to include your abstract after the title page.

Response Done s suggested

Comments Reviewer #1: Important note: This review pertains only to ‘statistical aspects’ of the study and so ‘clinical aspects’ [like medical importance, relevance of the study, ‘clinical significance and implication(s)’ of the whole study, etc.] are to be evaluated [should be assessed] separately/independently. Further please note that any ‘statistical review’ is generally done under the assumption that (such) study specific methodological [as well as execution] issues are perfectly taken care of by the investigator(s). This review is not an exception to that and so does not cover clinical aspects {however, seldom comments are made only if those issues are intimately / scientifically related & intermingle with ‘statistical aspects’ of the study}. Agreed that ‘statistical methods’ are used as just tools here, however, they are vital part of methodology [and so should be given due importance].

COMMENTS: The effect size is taken from a pilot study with total n=10 (i.e., 5 subjects in each group assuming 1:1 allocation ratio, which is too small to yield any useful/meaningful statistic) [Sample size section (line 163-7): The effect size estimate regarding the primary outcome MFIS, was obtained from a pilot study in which 10 participants underwent a similar protocol. The effect size for the differences between the groups was estimated to be 0.79. Accordingly, given a significance level of 5% and statistical power of 80%, 25 patients for each treatment arm were needed. To compensate for potential attrition, we increased the sample size by 20%.giving a total number of 30 patients per group]. The effect size used to estimate required sample size seems to be very large {unreasonably large}. In such (psychological therapy) studies [The objective of this study was to explore the added value effect of cognition-targeted exercise versus symptom-targeted exercise for MS fatigue], the effect size achieved/attained/seen is generally (in most cases) is of small to ‘medium’ level.

By referring to table-2 on page 158 of J. Cohen’s paper “A power primer” in Psychological Bulletin, 1992, vol.:112, pp 155-159 even for medium effect size you need n=64 per group (type-I error=0.05, power=80%).

{Please note that the ‘effect size’ assumed should have some valid basis (exact published reference needs to be quoted) &/or reasonable/realistic, else the study is very likely ‘not to be able to’ detect a difference despite its presence}

Response I totally agree that this pilot small sample may be not representative for actual differences and may leads to a lack of representativeness in the study sample

And accordingly we changed our design to Randomized pilot trial

And recalculate the sample size based on the published minimal clinical difference for the primary outcome 

Comments Note that ‘The general linear model with repeated measures’ is not a technique originally developed (for testing interaction terms it is definitely nice, however, not) for between groups comparison. Direct between groups comparison techniques are available. Remember that though the measures/tools used are appropriate [e.g. modified fatigue impact scale (MFIS), work and social adjustment scale, hospital anxiety and depression scale, perceived stress scale] most of them yield data that are in [at the most] ‘ordinal’ level of measurement [and not in ratio level of measurement for sure {as the score two times higher does not indicate presence of that parameter/phenomenon as double (for example, a Visual Analogue Scales VAS score or say ‘depression’ score)}]. Then application of suitable non-parametric test(s) is/are indicated/advisable [even if distribution may be ‘Gaussian’ (i.e. normal)]. Agreed that there is/are no non-parametric test(s)/technique(s) available to be used as alternative in all situation(s) [suitable / most desired/applicable], but should be used whenever/wherever they are available. [Assumptions of ‘general linear model with repeated measures’ are unlikely to be fulfilled making their use invalid]

Response To follow up and compare the effects of the 2 alternative treatments over 3 months, we examined the results using a generalized Estimation Equation (GEE) Model .

Page 9

Comments What is in lines 174-5 (we examined the results using 2 way repeated measures anova) is not reflected properly in tables [2 & 3]. Why there are two ‘P’ values [one each in column 4 & 5, row 4]. Application of repeated measures ANOVA yields only one ‘P’ (for each parameter/variable), in my knowledge. Then which test it is? If it is done after getting a significant ‘F’ by repeated measures ANOVA, then is not that “multiple testing” which requires adjustment in ‘P’? What the CI [in column 4 & 5, row 4] is for? Is it for difference in means? What is the interpretation? Moreover, one should use ‘to’ in/while presenting CI {example, [-5.9 -3.6] should be [-5.9 to -3.6]}.

Response This part was rewritten again

Comments It is well-known that while reporting [findings from] ‘Clinical Trial’ follow CONSORT guidelines. Even word ‘CONSORT’ or important items {like Random Sequence generation (Item 8a), Allocation concealment (Item 9)} of/in CONSORT checklist are not found [since your article type is ‘Clinical Trial’, you are supposed to cover these items in the report].

Response We added the required information in Page 5

Comments Two limitations of this study pointed out in lines 254-7 are not (unfortunately) the only ones. 

Response We added more relevant limitations 

Page 13

Comments Reviewer #2: Title

What did you need in the title

Discuss What do you mean here by added value?

The CBT used too much and published before, what was the new in your study

What is your question and answer here?

Response I explained the rationale behind this study in the introduction part 

Comments Abstract:

The background needs to be shortened.

Methods section is poorly framed. It has to be re-written.

Demographic profile of patients is not mentioned. P value are all .0005 – please check again

poorly Key words and not illustrative (type of technique or therapy you used )

Response Done as suggested

Comments Introduction:

1. Explain the rationale of the study. Please delete information unrelated to objective so that the section is short and sweet. Kindly focus on three elements of introduction.

a. What is known about the topic? (Background)

b. What is not known? (The research problem)

c. Why the study was done? (Justification)

2. Objective is not clear as mentioned above.

Write on

patients with multiple sclerosis

cognitive behavioral model of MS fatigue

Symptom-targeted Exercise

objective of the current study

Response Done as suggested

Comments Methods

Totally unclear

Rewrite by these consequences

Study design, setting, sample size, Participant (inclusion and exclusion criteria),

-Which stage of MS you select –mention if the patients have any symptoms or remission –the patient under treatment during the study (mention all with its side effect )

-Intervention (explain CBT)

- Comparison. Ethics and end point

Response I rewrote this part following these suggested arrangement and added more details for CBT 

Medications at enrolment and types of MS were summarized at table 2

Comments LINE 101-102

psychological disorders or any chronic illness that may affect their fatigue

discus what are you mean here

Response It is a typo error

I meant her psychiatric disorders. 

Page 5 line 120 

especially Previous cross-sectional studies have demonstrated a close association between chronic fatigue and psychiatric disorders.

Skapinakis P, Lewis G, Meltzer H. Clarifying the relationship between unexplained chronic fatigue and psychiatric morbidity: results from a community survey in Great Britain. Am J Psychiatry. 2000;157:1492–1498. [PubMed]

Lawrie SM, Pelosi AJ. Chronic fatigue syndrome in the community. Prevalence and associations. Br J Psychiatry. 1995;166:793–797

Comments LINE 103

A third person blinded to group allocation

Where were the 1st and 2nd person

Response The term "third person" refers to someone else, i.e., not the treatment provider or assessor .

And anyway I have changed to an independent person page 5

Line 123 To avoid any confusion

Comments Line 110- 111 (explain CBT )

The subjects in the experimental group received eight 50-minute sessions of weekly CBT based on van Kessel’s model

Response We added more details for CBT table 1

Comments CONSORT 2010 checklist need revision pages not accurate for each element

13a and 13b not covered in page 8

Page 8 include (data analysis and results )

Response Adjusted as suggested 

Comments Results

P value are all P<0.0005 in the 1st paragraph page 9 and 0.001 in the tables clarify please (why same results for all values )

Also in table 1 gender (numbers not clear )30-25 v 29-26

I think statistics need readjustment completely

Response We rewrote this part as suggested

Comments Discussion:

The discussion section needs to be written and described scientifically after correction needed for results

Response Done as suggested

Comments References

Rewrite again

Most reference are old like (5,8,13,14,18,20,30) and not completed (volumes and issues ) like (33,34,35,29,26,15,1)

Response Done as suggested

Comments Reviewer #3: This work explores a novel target for the treatment of fatigue in patients with MS, and I congratulate the authors for their efforts to carry out this trial.

As additional data, it would be interesting to know the proportion of patients with different MS phenotypes recruited in the study, and if there were differences in the response to therapy in the different clinical forms.

In addition, it would be interesting to know if the patients who intervened were under immunomodulatory treatment, and what type of treatment, since we know that interferons can cause fatigue

Finally, an English correction by a native speaker would be recommended.

Response Done as suggested

---

## [Decision Letter · Decision Letter 1]

7 Sep 2021

PONE-D-21-18476R1The added value of cognition-targeted exercise versus symptom-targeted exercise for multiple sclerosis fatigue: a randomized controlled pilot trialPLOS ONE

Dear Dr. Moustafa,

Thank you for submitting your manuscript to PLOS ONE. After careful consideration, we feel that it has merit but does not fully meet PLOS ONE’s publication criteria as it currently stands. Therefore, we invite you to submit a revised version of the manuscript that addresses the points raised during the review process.

We look forward to receiving your revised manuscript.

Kind regards,

Walid Kamal Abdelbasset, Ph.D.

Academic Editor

PLOS ONE

Journal Requirements:

Reviewers' comments:

Reviewer's Responses to Questions

**Comments to the Author**

1. If the authors have adequately addressed your comments raised in a previous round of review and you feel that this manuscript is now acceptable for publication, you may indicate that here to bypass the “Comments to the Author” section, enter your conflict of interest statement in the “Confidential to Editor” section, and submit your "Accept" recommendation.

Reviewer #1: (No Response)

Reviewer #2: All comments have been addressed

Reviewer #3: All comments have been addressed

2. Is the manuscript technically sound, and do the data support the conclusions?

Reviewer #1: (No Response)

Reviewer #2: Partly

Reviewer #3: Yes

3. Has the statistical analysis been performed appropriately and rigorously? 

Reviewer #1: (No Response)

Reviewer #2: I Don't Know

Reviewer #3: Yes

4. Have the authors made all data underlying the findings in their manuscript fully available?

Reviewer #1: (No Response)

Reviewer #2: Yes

Reviewer #3: Yes

5. Is the manuscript presented in an intelligible fashion and written in standard English?

Reviewer #1: (No Response)

Reviewer #2: Yes

Reviewer #3: Yes

6. Review Comments to the Author

Reviewer #1: COMMENTS: Not all of the comments made on earlier draft(s) by me are addressed satisfactorily [example: In response to my comments that ‘The general linear model with repeated measures’ is not a technique originally developed for testing the difference between groups, and that ‘assumptions of ‘general linear model are unlikely to be fulfilled making their use invalid’ your response is “To follow up and compare the effects of the 2 alternative treatments over 3 months, we examined the results using a generalized Estimation Equation (GEE)”]. In response to some other comment, you just said that “This part was rewritten again” but WHERE is not indicated.

In short, I am not very happy about the revision. Let the respected editor decide the future course.

Reviewer #2: -

Reviewer #3: THE CHANGES IN THE MANUSCRIPT WERE POSITIVE AND I CONSIDER THAT IT IS IN A CONDITION TO BE PUBLISHED

7. PLOS authors have the option to publish the peer review history of their article (what does this mean?). If published, this will include your full peer review and any attached files.

Reviewer #1: No

Reviewer #2: No

Reviewer #3: No

---

## [Author Response · Author response to Decision Letter 1]

15 Sep 2021

COMMENTS: Not all of the comments made on earlier draft(s) by me are addressed satisfactorily [example: In response to my comments that ‘The general linear model with repeated measures’ is not a technique originally developed for testing the difference between groups, and that ‘assumptions of ‘general linear model are unlikely to be fulfilled making their use invalid’ your response is “To follow up and compare the effects of the 2 alternative treatments over 3 months, we examined the results using a generalized Estimation Equation (GEE)”]. In response to some other comment, you just said that “This part was rewritten again” but WHERE is not indicated.

In short, I am not very happy about the revision. Let the respected editor decide the future course.

Response

At the beginning, I would like to make it clear that I completely agree with the reviewer and that this Generalized estimating equations (GEE) was chosen after consulting a biostatistician . This biostatistician was added to the manuscript based on the required modifications

In this study, We have selected the Generalized estimating equations (GEE) ,which considered as an extension of generalized linear models (GLM) because the major strength of GEE is that they do not require the correct specification of the multivariate distribution 

Especially the reviewer 1 has highlighted that a Generalized Linear Mixed Model (GLMM)require some parametric assumptions. And according to literature the Generalized estimating equations (GEE) are a nonparametric way to handle this issue (1)

(1) A Ziegler, M Vens -Generalized estimating equations . Methods of information in medicine, 2010 - thieme-connect.com

And accordingly, we have changed many sections in the manuscript 

Data analysis Page 9, line 109-201

Result, page 10, line 220-234

Abstract, line 43-44

reviewer comment 

What is in lines 174-5 (we examined the results using 2 way repeated measures anova) is not reflected properly in tables [2 & 3]. Why there are two ‘P’ values [one each in column 4 & 5, row 4]. Application of repeated measures ANOVA yields only one ‘P’ (for each parameter/variable), in my knowledge. Then which test it is? If it is done after getting a significant ‘F’ by repeated measures ANOVA, then is not that “multiple testing” which requires adjustment in ‘P’? What the CI [in column 4 & 5, row 4] is for? Is it for difference in means? What is the interpretation? Moreover, one should use ‘to’ in/while presenting CI {example, [-5.9 -3.6] should be [-5.9 to -3.6]}.

Response :

We have changed the statistical technique used in this study 

Instead of GLM we used Generalized estimating equations (GEE)

 and accordingly, we have changed the tables completely 

Now I believe the analysis technique is properly reflected in the table 

reviewer comment

Two limitations of this study pointed out in lines 254-7 are not (unfortunately) the only ones. 

Response 

We added more relevant limitations 

Page 13&14 lines 282 -289

"we propose several limitations of this study, pointing to necessary future research work on this topic. First, our project only included a short-term follow-up of 3-months after intervention was terminated. It is unclear how long the identified fatigue related changes would remain and whether patient compliance may also be an issue. Second, because we chose participants with a scoring of less than six in the expanded disability status scale to investigate, it is unknown how our study results might relate to people with MS who have higher levels of disability. Third, within the resource constraints of a pilot investigation, only a small sample size was feasible. While our study had statistically significant findings and raised the possibility of Cognition-Targeted Exercise CTE benefits, no reliable conclusions about mood effects of CTE can be drawn from such a small sample. As Button et al [43] emphasize, “a study with low statistical power reduces the likelihood that a statistically significant result reflects a true effect.” Thus, replication studies with larger sample sizes are critical to further evaluating any potential mood effects of this type of exercise . Despite of the above mentioned limitations, the present study can be regarded as a positive step toward finding and adopting non-pharmacological interventions to ameliorate fatigue levels in patients with Multiple Sclerosis"

---

## [Decision Letter · Decision Letter 2]

5 Oct 2021

The added value of cognition-targeted exercise versus symptom-targeted exercise for multiple sclerosis fatigue: a randomized controlled pilot trial

PONE-D-21-18476R2

Dear Dr. Moustafa,

We’re pleased to inform you that your manuscript has been judged scientifically suitable for publication and will be formally accepted for publication once it meets all outstanding technical requirements.

Kind regards,

Walid Kamal Abdelbasset, Ph.D.

Academic Editor

PLOS ONE

Reviewers' comments:

Reviewer's Responses to Questions

**Comments to the Author**

1. If the authors have adequately addressed your comments raised in a previous round of review and you feel that this manuscript is now acceptable for publication, you may indicate that here to bypass the “Comments to the Author” section, enter your conflict of interest statement in the “Confidential to Editor” section, and submit your "Accept" recommendation.

Reviewer #1: All comments have been addressed

2. Is the manuscript technically sound, and do the data support the conclusions?

Reviewer #1: (No Response)

3. Has the statistical analysis been performed appropriately and rigorously? 

Reviewer #1: (No Response)

4. Have the authors made all data underlying the findings in their manuscript fully available?

Reviewer #1: (No Response)

5. Is the manuscript presented in an intelligible fashion and written in standard English?

Reviewer #1: (No Response)

6. Review Comments to the Author

Reviewer #1: COMMENTS: All of the comments made on earlier draft(s) by me, were/are attended. Now the manuscript is acceptable.

7. PLOS authors have the option to publish the peer review history of their article (what does this mean?). If published, this will include your full peer review and any attached files.

Reviewer #1: **Yes: **Dr. Sanjeev Sarmukaddam

---

## [Editor Report · Acceptance letter]

28 Oct 2021

PONE-D-21-18476R2 

The added value of cognition-targeted exercise versus symptom-targeted exercise for multiple sclerosis fatigue: a randomized controlled pilot trial 

Dear Dr. Moustafa:

I'm pleased to inform you that your manuscript has been deemed suitable for publication in PLOS ONE. Congratulations! Your manuscript is now with our production department. 

Kind regards, 

on behalf of

Dr. Walid Kamal Abdelbasset 

Academic Editor

PLOS ONE